# The Perspectives of Programme Staff and Recipients on the Acceptability and Benefits of the Ward-Based Outreach Teams in a South African Province

**DOI:** 10.3390/healthcare8040464

**Published:** 2020-11-05

**Authors:** Cheryl Nelson, Sphiwe Madiba

**Affiliations:** Department of Public Health, School of Health Care Sciences, Sefako Makgatho Health Sciences University, Pretoria 0001, South Africa; cheryl.nelson.310@gmail.com

**Keywords:** re-engineering, primary health care, outreach teams, programme recipients, health benefits, acceptability, South Africa, rural

## Abstract

The re-engineering of primary health care (PHC) called for the establishment of ward-based outreach teams as a reform strategy to bridge the gap between health facilities and communities. The Nkangala district established ward-based outreach teams in 2012. We used process evaluation to assess the acceptability of the outreach teams from the perspectives of those involved in the implementation as well as the clients who are the recipients of the outreach services in order to describe how the programme benefits the recipients, the staff, and the health system. Data were collected through interviews with multiple data sources. A thematic analysis was done using NVivo 11. The outreach programme is acceptable to the recipients and staff. The acceptability translated into measurable benefits for the recipients and the health system. Health benefits included increased access to services, support for treatment adherence, and linkages to various sector departments for social support. Since the inception of outreach teams, the district has recorded low utilisation of PHC services and improved priority indicators such as immunisation coverage, early antenatal bookings, treatment adherence, TB cure rates, and decreased default rates. The positive effects of the outreach teams on indicators underscore the need to roll the programme out to all sub-districts.

## 1. Introduction

The National Department of Health in South Africa introduced the re-engineering of primary health care strategy as one of the reforms in a set of health system reforms in 2011. The goal was to address weaknesses in the system that resulted in the country only partially achieving the Millennium Development Goals related to maternal, child and infant mortality, HIV, and TB [1]. The re-engineering of primary health care strategy recommended, among a number of reforms, the adoption of a ward-based outreach team strategy to strengthen health promotion, identifying individuals and families at high risk of disease, and building links between households and health care facilities. The ward-based outreach team (WBOT) programme in South Africa is aimed at addressing the limitations previously experienced with community-based health services for vulnerable communities in priority settings like rural areas. It seeks to improve access to primary health care (PHC) services with the aim of improving health outcomes [2].

The key to achieving better health outcomes is a reform of South Africa’s health care system—specifically, moving from an individualised, passive, curative, and vertical system to a population-based, integrated, and proactive primary health care model [3]. The re-engineering of primary health care aims to support a preventive and health-promoting community-based PHC model using ward-based outreach teams [4]. In South Africa, wards are geopolitical subdivisions of metropolitan and local municipalities used for electoral purposes. They are the lowest political units of a municipality. The outreach teams play a critical role in extending PHC services to the community and household level and making health care accessible in terms of distance and information [5]. The WBOTs provide means to bridge the gap between health facilities and the communities they serve, a role which may be especially valuable in rural areas where individuals have to travel excessive distances to seek medical care [6,7,8].

Community healthcare workers (CHWs) are recognised as central to the re-engineering of primary health care because of their capacity to work closely with vulnerable communities and individuals. As such, the WBOTs in South Africa are primarily constituted of CHWs under the supervision of an operational team leader or outreach team leader, usually a nurse. Each team consists of up to six CHWs, is responsible for a defined number of households, and is accountable to a local health facility. Each outreach team serve a population of about 6000 with at least 1000 to 1500 households (depending on density, the burden of disease in the area, and geography). Each CHW is responsible for 250 households [2].

The role of the CHWs in the outreach teams is mainly to strengthen the interface between the households and health facility services by linking communities to the healthcare system [1]. They offer community-based healthcare and social support to complement rather than replace the more specialised services of the healthcare system [9]. In South Africa, CHWs are required to have completed the 12th grade and 69 days of skills training; they receive a 10-day orientation followed by practicals and in-service training [10]. The National Department of Health defines and standardises the CHW’s scope of work, remuneration packages, and other working conditions. CHW’s scope of work include health promotion, primary prevention of disease, healthy behaviour counselling, treatment adherence counselling, secondary disease prevention through basic screening with appropriate referral, and basic therapeutic, rehabilitative, and palliative care services to vulnerable communities [11]. CHWs are remunerated through a system of stipend payments through intermediaries. Remuneration levels are highly variable and inconsistent across the country; CHWs can earn in the range of R1800–R3500 (USD $150–290) per month [12].

The outreach teams are linked to PHC facilities and leverage facility-based resources to deliver their services. The PHC facility provides support to the outreach teams and receives referrals of patients they see and diagnose in the community [7]. The main functions of the outreach teams are to promote good health and prevent illness through a variety of interventions based on the concept of a healthy community, a healthy family, a healthy individual, and a healthy environment. The community outreach team leader is responsible for ensuring that the team’s work is targeted and linked to service delivery targets and that the team is adequately supported and supervised [2]. The CHWs in the outreach teams constitute the first point of contact on health-related issues and play a significant role in the delivery of health services to the community.

South Africa faces a formidable burden of illness, and the high burden of the HIV and AIDS epidemic coincides with high burden of tuberculosis, high burden of non-communicable disease, high levels of violence and injuries, and high maternal and child mortality [13]. Hypertension and diabetes are key components of the country’s rapidly growing burden of non-communicable diseases. A recent national demographic and health survey found that 44–46% of South African adults were hypertensive, and 31% of men and 68% of women were considered overweight or obese [14]. Therefore, the role of the outreach teams includes addressing HIV/TB, maternal-child health, and chronic non-communicable diseases [12].

There is a growing body of evidence to suggest the contention that CHW interventions provide a major benefit to communities [15]. If it is implemented well, the WBOT programme has the potential to address limitations experienced previously with community-based health services, to improve access to PHC services, and, subsequently, to improve health outcomes. The introduction of outreach teams could play a role in lightening the burden associated with pursuing healthcare in health facilities and the availability of referral pathways, benefitting households [5]. There is also evidence that WBOTs have a positive impact on healthcare [16]. In a study conducted in KwaZulu Natal, household members expressed an appreciation of the way in which the WBOT services have brought health care closer to the people [5].

The WBOT programme was launched in the Mpumalanga province in 2011, and Nkangala district established its first WBOT teams in three sub-districts in 2012. It is important to understand the nature of the programmes to inform the roll out of the WBOTs in other sub-districts in the province. Process evaluation assumes greater importance in the case of large, complex community-based intervention such as the WBOT programme, which deliver multiples, non-standardised interventions tailored to specific communities [17]. For the WBOTs to contribute meaningfully to health reforms in South Africa, they require careful planning and extensive support by communities and other key stakeholders with the communities. In addition, better health outcomes related to the WBOTs should involve a comprehensive understanding of a far greater array of community and health system factors [18].

Given the pivotal role that the outreach teams play at the community level, it is important to determine if the community perceives them as acceptable. Research shows that barriers against and facilitators in community-based services are related mainly to programme acceptability, appropriateness, and credibility [19]. Moreover, the main emphasis in the implementation of the outreach programme is on improving health outcomes. However, there is limited information available in South Africa on user perceptions of the services provided by WBOT programmes in rural households, and there has not been a previous evaluation of the implementation of the programme in the Nkangala district. It is important that countries introducing outreach teams institute robust evaluation studies as they launch their programmes, and that they include studies conducted from the perspective of household users [20]. The evaluation of the implementation of the programme in the Nkangala district will inform the roll out of the WBOTs in other sub-districts in the Mpumalanga province.

The purpose of the evaluation was to assess how the ward-based outreach team programme is being implemented by managers, team leaders, and CHWs, and to assess the acceptability of the programme.

The process evaluation answered the following questions:How is the primary health care ward-based outreach programme currently being implemented in the Nkangala district?How acceptable is the ward-based outreach programme to managers, community health workers, and clients in the district?What are the barriers and facilitators for the implementation of the ward-based outreach programme?What are the recommendations of managers, CHWs, and clients to improve the implementation of the WBOT programme in the district?

This paper assesses the acceptability of the WBOT programme in the Nkangala district from the perspectives of those involved in the implementation as well as the clients who are the recipients of the outreach services, and describes how the outreach programme benefits the recipients, the CHWs, the outreach team leaders, and the health system.

## 2. Materials and Methods

### 2.1. Design

The framework for this evaluation was informed by the logic model framework developed by the Centers for Disease Control and Prevention. The logic model framework informs the evaluation practice and is composed of six steps, but in a process evaluation, the focus is on the first three segments of the logic model (inputs, activities, and outputs) and how they work together. The framework was used as a theory of change to present and share the understanding of the relationships among the programme inputs or resources available to implement the programme, the activities, and their links to the anticipated outcomes of WBOTs [21] (Figure 1).

We also integrated the three domains of evaluation recommended by the Medical Research Council guidance on process evaluation, namely implementation, intervention mechanisms, and contextual factors [22]. This paper focusses on the third domain, which involves the identification of contextual elements that positively or negatively affect implementation and outcomes. The updated guidelines stress the relevance of taking the contextual factors associated with variations in implementation, intervention mechanisms, and outcomes into account [22]. The context of the programme includes internal and external factors such as the organisation-related factors, characteristics of the programme, characteristics of HCWs involved, and the context in which the programme is taking place. The evaluation will assess whether these contextual factors have an influence on the implementation of the programme. The external context in which WBOT is taking place also includes the uptake and acceptability of the program by the community. For example, the evaluation will assess whether cultural beliefs and practices in communities where the outreach program is being implemented will influence the uptake and acceptability of the outreach teams.

### 2.2. Evaluation Sites

The evaluation was conducted in three sub-districts implementing the WBOT programme in the Nkangala district, one of the three health districts in Mpumalanga province in South Africa. The district is rural, and it is the second largest populated district in the province with a population estimated at 1,433,047. The three sub-districts were selected on the basis that they had commenced the implementation of the WBOT programme in 2012.

### 2.3. Sampling and Data Sources 

The data sources consisted of healthcare workers involved in the outreach programme and included the provincial and district coordinators, the district manager, the facility supervisors, the facility managers, the team leaders, CHWs, and clients who had been programme recipients. Only those directly involved in the implementation of the programme and who could provide rich answers to the evaluation questions were included in the study (Table 1). The evaluation excluded managers, supervisors, and CHWs who had been performing WBOT-related activities for fewer than six months. Concerning clients, the evaluation included only adult clients who had been programme recipients for over six months. The sample for community members to participate in focus group discussions (FGDs) was purposeful. Sampling was done with the assistance of the community health workers who identified households and household members who meet the inclusion criteria. 

### 2.4. Data Collection

Data were collected during November and December 2016 following the receipt of ethics approval from Sefako Makgatho University’s Research and Ethics Committee (SMUREC/H/157/2016: PG), the receipt of all relevant permits from the Mpumalanga Province Research and Ethics Committee, and the support of the management of the Nkangala district.

The data collection methods for this evaluation were in-depth interviews with multiple data sources—district and facility managers, outreach team leaders (OTLs), and CHWs—to evaluate the implementation of the outreach programme. The interview sites for the provincial and district managers were the district offices, the sub-district offices for the supervisors, and PHC centres for the facility managers, the OTLs, and CHWs.

A semi-structured interview guide was developed to address three components of a process evaluation to assess how the ward-based outreach team programme was being implemented by managers, team leaders, and community health workers. To that effect, the perceptions of HCWs and CHWs were sought to assess the acceptability of the WBOT programme in the Nkangala district as well as how the programme benefitted the programme staff and recipients. Acceptability refers to determining how well an intervention will be received by the target population and the extent to which the intervention meet the needs of the target population and the organisational setting. The data sources were asked, among other questions, (1) how satisfied they were with the way the outreach team activities were carried out, (2) how acceptable the outreach program was, and (3) the benefits of the outreach programme to the community, the program staff, and the health system. They were also asked to make recommendations to improve the implementation of the outreach programme.

A focus group guide was used to conduct FGDs with clients who were recipients of the outreach programme. Each group had an average of 10 participants who were programme recipients for over six months. The focus groups were comprised of male, female, young, and elderly community members with and without chronic diseases. The guide elicited information on the perceptions of the clients on the acceptability of the outreach teams and their views on how the programme benefitted their households and the community at large. They were further asked to make recommendations on how to improve the implementation of the programme. Data collection was ceased after six focus groups had been conducted, as data saturation was achieved. Saturation was considered to have been achieved once no new information was being contributed to the understanding of the key evaluation questions or the implementation of the outreach team programme [23]. The sites for the focus groups were the clinics in the sub-district where the WBOT programme was being implemented.

The interviews and FGDs were conducted by the lead author and trained field workers with experience in qualitative studies. The field workers underwent a day’s training on the evaluation protocol prior to the data collection. Thirty-one interviews were conducted in English, and each interview lasted for about 60 min. Two focus groups were conducted per sub-district. The discussions were conducted in the local languages commonly spoken in the three sub-districts (Sepedi and IsiZulu). Digital voice recorders were used to record the interviews, with the consent of the participants. Signed, informed consent was obtained from all participants prior to data collection, and they were assured of the confidentiality and anonymity of the process. The community members and community health care workers were offered transportation to come to the interview venues. At the end of the interviews and focus groups, refreshments were provided.

### 2.5. Data Analysis

The interviews and FGDs were recorded, transcribed verbatim from IsiZulu and Sepedi, later translated into English by the field workers, and then analysed using the framework analysis approach. Both authors were involved in data analysis; they are fluent in English and the local languages and have a full understanding of the local culture and customs. Following the researchers’ repeated readings of the transcripts to familiarise themselves with and immerse themselves in the data, initial codes were identified from the data. A thematic analysis was done and a priori themes related to the evaluation research questions and the four-level framework defined by Stevenson and Ong [24] provided a starting point for the development of these codes. After the authors had familiarised themselves with the data and the initial codes had been discussed and agreed upon, all transcripts were coded using the NVivo 11 qualitative data analysis software [25]. The authors held regular meetings to continue discussing the codes in order to develop a shared understanding of them, and to refine and finalise the coding framework. The codes were further discussed and merged into emerging themes, further themes were added and refined, and barriers and facilitators were verified.

We triangulated different data sources to enhance the credibility of the data by exploring the perspectives of the CHWs, different categories of health professionals involved in the outreach programme, and programme recipients [26]. Multiple data sources responded to the same question on the acceptability of the outreach teams to the program staff and the community and on how the outreach teams benefitted the program staff and the community. We also used two methods of data collection—interviews with program staff and focus group discussions with programme recipients. The findings and conclusions drawn are based on triangulated data from multiple data sources. Rigour was further enhanced using an efficient electronic recorder to facilitate verbatim transcription, and thus to ensure that the views of the participants were accurately represented. Data quality checks were conducted during the transcribing of the scripts, the coding, and the development of the themes. The evaluator provided a full description of the context of the evaluation by keeping an audit trail that described all the procedures and processes followed during the conduct of the evaluation [27].

## 3. Results

### 3.1. Description of the Study Sample

The sources of the evaluation data included the provincial and the district WBOT coordinators, the district manager, two supervisors, three facility managers, eleven OTLs, and fourteen CHWs. The characteristics of the data sources included in the study are shown in Table 2. Most were in the 30–40 years age group and were female.

### 3.2. Themes

Analysis revealed two main themes and several sub-themes that reflect the benefits and acceptability of the WBOTs as perceived by those involved in the implementation as well as the programme recipients (Figure 2).

#### 3.2.1. Satisfaction with the Outreach Teams

The acceptability of the outreach teams to household community members, CHWs, OTLs, and managers was determined.

##### The Perspectives of Recipients of the Programme 

The community members who were the recipients of the outreach services stated that the outreach teams were accepted well in the community. There were various reasons why they felt the programme was being received well by the community.


*“They are not harsh; they are so down to Earth and talk to you nicely. They ask for your permission. If it were not for them, we wouldn’t be where we are today. Even if there is someone who refuses to test, they talk to him or her nicely; in the end they agree to go and test. They help us a lot in our families since they arrived. They have this way of talking to you. I do not know if that is how they have been trained. They are so polite in a way that you will feel important.”*
CM


*“Their visit made me feel better…I had already stopped taking my medication because the staff at the clinic were shouting at me. So, when they came to visit me, I explained what had happened at the clinic. They gave me a referral letter to take with when I go to the clinic. They assisted me.”*
CM


*“The other thing is when you have a problem and you need them, they do come to you and assist with your problem.”*
CM

In contrast, a few mentioned that some community members were not receptive of the outreach teams.


*“The community does accept it, but not the whole community; at times in other houses, when they arrive and greet the people, they throw them out in a nasty manner. Some do accept them, but not all the community.”*
CM


*“That is why at times they are scared to go to some homes, because the people become nasty and become impatient.”*
CM


*“We also hide when they come, we tell our children that they must say we are sick obvious they will be dishearten and go back. The fact is that we do turn them back.”*
CM


*“They should change their system if they can visit us monthly and ask how we are doing; they should not come only in December. They should come every month and check if there is no one who is sick. They should not come at the end of the year only and not know what is happening during the whole year.”*
CM


*“They are not many, they are very few, and they knock off at twelve. They are few and this area is big. So, they only see you once a year or after three months; it is impossible for them to see you every month.”*


##### The Perspectives of the CHWs

Most of the CHWs were of the view that the community is appreciative of the outreach teams. They stated that most households have indicated that they would like to see the outreach teams regularly. They further explained that most households follow the instructions of the outreach teams, and that by doing so, they get the help that they need.


*“The community trust us. When they see us, they see help even if we don’t have anything with us. By just entering their households with a smile, they feel healed already.”*
CHW


*“The community appreciate the programme a lot. They wish to be with us every day, but it is impossible because we work with many homes.”*
CHW

Some of the CHWs indicated that in some of the households, they were not accepted by household members. They were also not received when they did not have uniforms and nametags as a form of identification.


*“In some homes, it’s difficult; they say, “No, we are busy, you are taking our time.” Even though it’s like that, we are forced to sit down and have a conversation with them.”*
CHW


*“Some people in the households used to chase us away; others unleash their dogs on us so they could bite us.”*
CHW

They also said that having a good relationship with the households and key community stakeholders increased the acceptance of outreach teams.


*“The chiefs and community development workers support the programme.”*
CHW

On the other hand, on a personal level, some of the CHWs expressed their dissatisfaction with the programme.


*“At first, I loved this, and I was having the strength, but now I no longer have the strength. I am thinking of getting another job in town, just cleaning for someone instead.”*
CHW


*“They cannot even give us a decent salary, with that R1200.00 I am expected to buy clothes to wear when I am in the community. They will not allow me not to be in uniform.”*
CHW


*“It hurts a lot because sometimes when you come back from the field, you are tired, and you don’t know what you are going to eat. But just because we’re aware of what we are dealing with in the field, we don’t have a choice and we are willing to help these people so that they can have a good life.”*



*“The referral system is not working…it’s bad, we refer patients to the clinic, but the nurses do not refer back to us. They undermine us. They say we are not trained, we don’t know this job, and then they ignore us.”*
CHW

##### The Perspectives of Health Professionals

The programme staff felt that the programme reduced the burden on clinics by improving access to services at community level. The facility managers were satisfied with the programme because they had noticed an improvement in the facility health indicators.


*“If there were ratings, I would give them ten out of ten. I would give the OTL nine-and-a-half out of ten because she was really performing. Most of the facility outcomes were at the level that I needed.”*
Facility manager


*“I think I am satisfied because the outreach teams assist a lot in tracing chronic patients who have defaulted.”*
Supervisor


*“I am so satisfied because in the field, I know when they are there, they will make sure everything is done perfectly.”*
Facility manager

The OTLs were very satisfied with the programme and gave various reasons for their satisfaction.


*“I am happy because the immunisation statistics is very good. Very few defaulters have valid reasons to default the treatment. So, we have achieved a lot. Even the TB rate and pregnancy rate has dropped.”*
OTL

The OTLs said that there is general satisfaction about the programme among different stakeholders who commended them about the programme and advocated for establishing more outreach teams.


*“The community is now used to the outreach programme and there are many communities that do not have this outreach programme and they wish to have it in their communities.”*
OTL

The provincial and district management were satisfied with the outreach teams. They expressed satisfaction with the WBOT programme and said that it was pleasing to see how the programme had advanced from the time of its inception.


*“I’m actually satisfied, you know. It becomes so self-fulfilling when you see something that was started from scratch…and we had to put systems in place to make sure that we start implementing the programme which was never implemented anywhere in the country.”*
District manager

The evaluation also found that consultation with relevant stakeholders including traditional health practitioners, ward counsellors, and ward committees was key in the acceptance of the outreach programme. The outreach team works with sector departments to arrange meetings with relevant stakeholders.


*“We have community meetings with the counsellors to discuss how we can work together.”*
OTL

However, some facility managers were not satisfied and some OTLs expressed dissatisfaction with how the National Department treats the CHWs.


*“I am not satisfied because if the OTL is in the clinic doing admin work, I feel she can also help with clinic-related work.”*
Facility manager


*“I am not that satisfied because basic things like uniform for CHWs is not that expensive, these are things that the department can do. The same applies to the CHWs that we are working with, to absorb them in the system. I feel it’s something very easy.”*
OTL

#### 3.2.2. Benefits of the Outreach Teams

##### Benefits to the Communities

The community members reported several benefits of the programme.


*“I don’t like drinking my pills, but when the outreach teams arrived in my household, I received them. They talked to me and put me in a right place, and I saw that taking medication is good for me.”*
CM


*“They help us a lot by coming to visit us in our homes and take good care of us. There are sick people who are scared to go to the clinic because the nurses do not approach them well. So, if they come to households, many people get well, because they are able to talk to them about anything. Even secrets, we are able to share with them.”*
CM


*“She (OTL) stays in our vicinity, even at night, this woman wakes up, she does not have a problem, when you knock, she wakes up, you tell her your challenge and she helps you. I have never seen a person like her at the clinic, I come at night and stay for a long time, but she always accompanies me.”*
CM

They further stated that they benefitted from the programme because the outreach teams bring chronic medication to the households for sick people that are staying alone.


*“If you are living with someone who is sick in the household and you cannot leave that person alone, they are able to get medication on your behalf.”*
CM


*“When a person is sick and bed-ridden, they take the person to the clinic and get treatment for them. They help a lot.”*
CM


*“I once defaulted treatment, so they came and encouraged me to go back and start my treatment again. They found me in a horrible state, and I was very weak. They encouraged me, they helped me, and I now adhere to my treatment.”*
CM

The community benefitted from the collaboration between the outreach teams and other sector departments such as the Departments of Social Development, Basic Education, Human Settlement, Internal Affairs, the Social Security Agency of South Africa (SASSA), and centres for food parcels. The community also benefitted from the provision of food parcels.


*“They collect food parcels for children. When the children eat, they ask them about their families, and they do a follow-up, and they go to the homesteads to see conditions of those families.”*
Community member


*“Sometimes, there are orphans who don’t have birth certificates and do not receive grant. We have a centre where we are referring them to go and eat. We have vulnerable and child-headed households who are not getting grant. We send them again to SASSA to get food packages.”*
CHW

The CHWs indicated that the communities they serve show appreciation. The CHWs outlined the benefits of the programme, particularly for households. They reported that the community benefits from having health services brought to their households. They indicated that the communities consult them anytime when there is a need.


*“The community appreciates what the Department of Health does for them as they bring the health services to their households, because the community appreciates us.”*
CHW


*“The WBOT help with health issues so that the people shouldn’t go long distances to the clinic. Like our ward is very broad and they walk distances they are very far to reach the clinic so we are able to go to them.”*
CHW

##### Benefits to CHWs and OTLs

The CHWs reported that the training they had received had empowered them, even though it was not adequate.


*“As far as I’m concerned, I have learned a lot; I am now independent as far as my work is concerned, but because I have been given a team leader, I do need her all the time.”*
CHW

The OTLs reported that they now have a better understanding of the challenges that communities face, especially when they are sent from pillar to post by the different sectors of the departments.


*“You know, I’ve understood one thing ever since I worked outside of the clinic. Since I became part of the outreach teams, I saw that it is very easy to turn somebody away when you are at the clinic, when you don’t know where these people are coming from.”*
OTL


*“I mean, you learn more about your job description as a team leader, what you are expected to do as a team leader, and how to attend to other challenges, where to report to as a team leader, as well as the tools you are going to use as a team leader.”*
OTL

##### Benefits to the Service Provision

The facility managers felt that the clinic would not survive without the outreach teams. They noted an improvement in the priority indicators.


*“Through the WBOT, we are able to increase our priority indicators. Some of the mothers do not send their children for immunisation in the clinic. So, the outreach teams go out and finds these children and then we help them catch up with the immunisation. That helps to increase the statistics of the clinic. They are also able to find those who defaulted their treatment.”*
Facility manager


*“Since the programme started, we are getting more pregnant women and they are booking early for ANC. We also do not have more defaulters as before, and on the TB cases, we ask them to go trace patients who are defaulting, so the defaulter rate has been reduced.”*
Facility manager


*“Without the outreach team, the clinic won’t survive because even if they introduce vitamin A, the outreach team will go to the crèche and help the clinic by giving the children de-worming and vitamin A.”*
Facility manager

District management reported that the low PHC utilisation rate could be attributed to the implementation of the WBOTs in that the burden of disease is somehow reduced.


*“The fact is that the PHC utilisation rate, which is supposed to be around 3.5, but it is now around 2.7 and 1.8. Our view and analysis is that it appears as if the implementation of the WBOTs may have played a role. We don’t see a lot of patients at PHC level any more. This suggest that maybe the burden of diseases is reduced or the pressure put on PHC facilities is reduced.”*
District manager

They also reported a slight improvement in the indicators of facilities implementing the WBOT programme.


*“We were looking mostly at the indicators, facility indicators like ANC visits at twenty weeks, immunisation coverage, and post-natal visits within six days, vitamin A, and deworming. So, when you look at the DHIS and the facilities that are being supported by the ward-based outreach teams, there is a slight of improvement of these indictors.”*
District manager


*“Currently, we have seen an improvement in some of the priority indicators like the TB cure rate and the improvement in the early bookings for women and reduction on the defaulter rates on TB treatment, which we said it may be due to support of the WBOTs.”*
Provincial WBOT manager

## 4. Discussion

The evaluation established that the WBOT programme is acceptable to the recipients and key stakeholders which include, among others, households, district hospitals, clinics, schools, early learning centres, sector departments, police services, local chiefs, traditional healers, and local political councillors in the communities where the evaluation was conducted. The CHWs, OTLs, and facility managers shared the same sentiments about the acceptability of the outreach teams in the communities where the WBOT is being implemented. Findings from other provinces show that the WBOTs have been accepted well by key health system stakeholders in many parts of the country [7,28,29].

The evaluation established that having committed and motivated outreach teams and good attributes of CHWs is important in the implementation and acceptance of the outreach programme. The CHWs were patient and tolerant of insults and being denied access to households, and they dedicated their time to the communities to educate them about the outreach programme until both they and the programme had been accepted. We further learned from the evaluation that the respect the outreach teams accorded to the cultural beliefs and practices of households played a critical role in the uptake and acceptability of the outreach. The ability of CHWs to work in the specific cultural contexts affecting their patients underscored the value of the CHWs and their local knowledge of their communities [30].

There is evidence that the collaboration with different stakeholders in the community and sector departments has led to the acceptability of the WBOTs. The outreach teams collaborated with various sector departments to organise grants for disabled people, to address issues such as child neglect and child-headed households, and to assist with the registration of births. As being among their functions, the OTLs visited schools and crèches, working hand in hand with the DoE and ECD centres. Other collaborations were with centres that distribute food, local municipalities, war rooms, ward committees, and ward councillors. 

In a similar manner, the outreach teams in other provinces have engaged with various sectors, participated in intersectoral war rooms, and worked with local political structures [7,29,31]. The OTLs played a crucial role in advancing the acceptance of the outreach teams by the households and key stakeholders. Additionally, the outreach teams were supported by key stakeholders such as the chiefs of the respective wards. Lastly, ward councillors and community leaders were the key sources of support for the outreach teams, which facilitated access to households [29,30].

In the current evaluation and in others, the household members who were recipients of the outreach services expressed their satisfaction with the outreach teams and their appreciation of the services they provided [29] and described various aspects of it that they appreciated. They were appreciative of the approach of the outreach teams, who helped restore their trust in the clinics. Some of them had stopped taking their chronic medication because of their dissatisfaction with the clinics, but the interaction with an outreach team helped them restart treatment. They were also satisfied with the level of confidentiality displayed by the outreach teams. They were appreciative of the fact that the outreach teams collect their medication and accompany them to the clinics for further consultation. A study conducted in KwaZulu Natal reported that household members considered the outreach teams to be valued resources which bring services closer to the community [5]. In the current study, as in others, household members felt empowered to take control of their own health care and to take their medication as prescribed [32,33].

In contrast, some of the household members were not satisfied with the frequency of the visits by the outreach teams. They indicated that the outreach teams only visited their households once in 12 months because they are few and lack transport to reach many households. They were dissatisfied with the poor ambulance services and the fact that the ambulances are not linked to the outreach teams to make it easy for households in need to access them. Similarly, some of the CHWs were not satisfied with the WBOT programme. They expressed dissatisfaction with the remuneration, which included the amount of money they received as a stipend and the method of payment of the stipend. An evaluation of the outreach teams in the North West province showed that CHWs were not satisfied with their working conditions [29]. CHWs were dissatisfied with the relationship with the nursing staff in the facilities. They indicated that the nurses’ attitudes and ignorance of their referrals makes them feel very disheartened. In the Northern Cape, CHWs were dissatisfied with the support received from the healthcare system and cited the high workload, a lack of resources, and the open contracts that generate anxieties over their future [33].

On the other hand, CHWs indicated that they had gained more knowledge and the confidence to perform their job independently from the supervision they received. However, they were concerned or dissatisfied with the scope of their work given the vast experience they had working with communities. They indicated that their role should be expanded and that they should be allowed to do additional clinical tasks such as measuring the blood pressure of their clients. Findings from an Eastern Cape study indicated that the CHWs were satisfied with their outreach work and found it meaningful [31]. Furthermore, being associated with medical knowledge can be empowering and a source of inspiration for the CHWs [33].

A rapid assessment of outreach teams in the North West province indicated that facility managers expressed varying levels of support for the WBOTs programme [28]. Similarly, in the current evaluation, some facility managers found the recruitment of OTLs from the clinic unacceptable, given the extreme shortage of professional nurses in clinics. They felt that the OTLs should also perform clinical work and not focus only on their outreach duties [34]. While in general, the OTLs find the WBOTs acceptable, they were dissatisfied with the high workload, the lack of material resources, office space, and transport to aid in the performance of their outreach duties.

The community is the main beneficiary of the outreach teams, but the evaluation highlighted the benefits for the CHWs, the OTLs, and the health system. The health benefits for the community included increased access to services, adherence to treatment, and support for ill and bedridden patients, and the provision of consumables like diapers for disabled patients. The community further benefitted from the collaboration between the outreach team and other sector departments. The collaboration led to the solving of problems related to low-cost houses, access to food parcels, access to social grants, applications for birth certificates, applications for identity documents, and the enrolment of school-going children at school. One of the greatest benefits of the outreach teams to the community is the availability of CHWs to do house calls as and when the need arises.

There are indications that WBOTs have made meaningful contributions to the health system [29]. The benefits include improvement in the health indicators such as increased immunisation coverage for children under one year, improvement in the TB defaulter rate, improvement in the TB cure rate, improvement in treatment adherence, antenatal booking before 12 weeks, and an increase in the number of children receiving vitamin A. Similarly, results from reviews of CHW programmes found that they make important contributions to improving health outcomes [35]. An evaluation of the WBOT programme in the North West province found that facilities linked to outreach teams appeared to perform better than facilities without them on some indicators [34]. An immunisation cover rate of above the 90% target and a decrease in the TB defaulter rate from 12% to 77% were reported [29].

A spin-off benefit of the outreach programme is the low PHC utilisation rate, although the quantification of this observation is uncertain. In addition, the health system has a better understanding of the health challenges that are faced by communities. This will inform the further development of the interventions made by the outreach teams and strategies for stakeholder engagement for the rollout of the WBOT programme in other districts in the province. Khuzwayo and Moshabela [32] attributed the reduction in PHC utilisation to the provision of care and support by outreach teams mainly in the form of household-level care, which reduces the need for clinic referrals and frequent clinic visits.

A key limitation of this study is that sampling for household members for participation in focus group discussion was done with the assistance of the community health workers who identified members in households that meet the inclusion criteria. It is possible that they could have sampled from those households who were satisfied with the outreach teams. This might have influenced their responses to the acceptability of the outreach teams. Another limitation is that we did not collect demographic characteristics of the household members to give their detailed description; we depended on data collected during the focus groups. Nevertheless, the sample consisted of a heterogeneous group that varied in age, gender, and health-related status. Member checking is often recommended as a method to evaluate validity in qualitative research, but the investigators did not use member checking to ensure that the findings are credibility. However, the findings are based on triangulated data from multiple data sources; triangulation is a strategy used to attain credibility.

## 5. Conclusions

The evaluation found acceptability of the WBOT programme among managers, OTLs, CHWs, and the clients who are the recipients of the programme in the evaluation sites in the Nkangala district. The acceptability of the WBOT programme translated into measurable benefits for the recipients of the programme, the CHWs, the OTLs, and the health system.

The Provincial and District Management, the facility managers, and the OTLs noted a decrease in default rates and an improvement in the immunisation coverage and in other health indicators such as an increased adherence to treatment and early bookings among pregnant women since the inception of the programme. The positive effects on the WBOTs on some of the primary care indicators underscore the need to strengthen the currents outreach teams but also to roll out the programme to other sub-districts in the Nkangala district.

The health benefits for the community members included increased access to services, support for treatment adherence, and support for ill and bedridden patients. There were also social benefits such as linkages to other departments for access to food parcels, access to social grants, and applications for birth certificates and identity documents, which were made through the intervention of the outreach teams. Though the results are from a process evaluation, they have implications for the strengthening of PHC and improving access to quality health care for all.

On the background of the early history of the WBOT programme, the district management stressed that the programme is cost-effective in that, in the short period since its inception, the district has seen a low utilisation of PHC services and improved priority indicators such as the immunisation coverage. This suggest that rolling WBOTs out to other sub-districts and the province will yield positive health outcomes at a cost that the province can afford.

## Figures and Tables

**Figure 1 healthcare-08-00464-f001:**
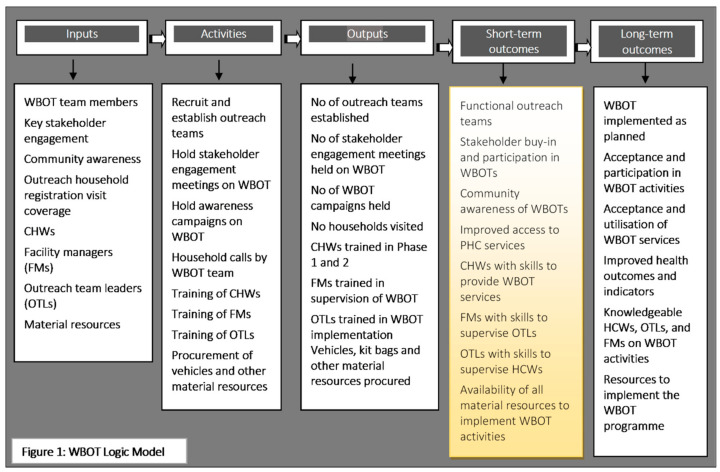
Ward-based outreach team (WBOT) logic model.

**Figure 2 healthcare-08-00464-f002:**
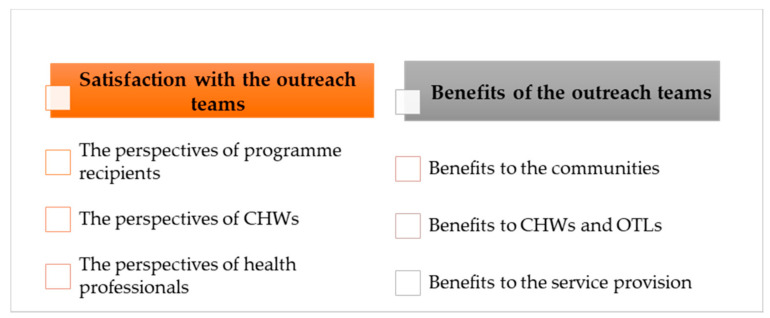
Themes.

**Table 1 healthcare-08-00464-t001:** List of data sources and evaluation sites.

Evaluation Site	Data Source	Population	Sample
Provincial Office	Coordinator	1	1
Nkangala District Office	District managers	1	1
Nkangala District Office	Coordinator	1	1
Dr JS Moroka Sub-District	Supervisors	2	1
	Facility managers	4	1
	Team leaders	4	4
	Community health workers	44	6
	Community members (clients)		2 FGDs
Thembisile Sub-District	Supervisors	2	1
	Facility managers	4	1
	Team leaders (OTLs)	4	4
	Community health workers (CHWs)	35	6
	Community members (clients)		2 FGDs
Emalahleni Sub-District	Supervisors	2	1
	Facility managers	2	1
	Team leaders (OTLs)	2	2
	Community health worker (CHWs)s	24	6
	Community members (clients)		2 FGDs

**Table 2 healthcare-08-00464-t002:** Data sources.

Participants	Number	Gender	Age
Provincial coordinator	1	Female	50
District manager	1	Male	51
District coordinator	1	Female	37
Supervisors	3	Females	45–58
Facility managers	3	1 Male, 2 females	45–55
Team leaders (OTLs)	10	2 Males, 9 females	40–59
Community health workers (CHWs)	18	Females	30–55

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
