# Peer review of "The Perspectives of Programme Staff and Recipients on the Acceptability and Benefits of the Ward-Based Outreach Teams in a South African Province"

_healthcare, 2020, doi:10.3390/healthcare8040464_

Round 1

Reviewer 1 Report

Quotes: should not be CM for community member or OTL, also include gender, date, age of the interviewer.

In fact, it would make the quotes stronger to use gender, date, and age.... as opposed using CHW.

Write up unnecessary acronyms: RPHC, NDoH, MDG, WBOT, CHW, MRC, FGDs, etc... there are just too many acronyms and make reading the manuscript almost impossible without having to re-read and re-re-read.

Results - need some clarity. Seems very rosy. In other words, too subjective.

Author Response

Response to reviewer comments

We thanks the reviewer for the comments and suggestions on our manuscript. We have addressed the comments to the best of our abilities, in the document; the response is highlighted in a light blue font

Comments and Suggestions for Authors

Quotes: should not be CM for community member or OTL, also include gender, date, age of the interviewer. In fact, it would make the quotes stronger to use gender, date, and age.... as opposed using CHW.

Response: We changed community member to CM, we however do not agree with the reviewer to add so much descriptors to the quotes because the aim of various reasons, one being that the aim of the evaluation was capture changes in implementation and experiences of the intervention without comparing the responses from the multiple data sources using demographic data. We provided a summary of the description of the data sources in table 2.

Write up unnecessary acronyms: RPHC, NDoH, MDG, WBOT, CHW, MRC, FGDs, etc. There are just too many acronyms and make reading the manuscript almost impossible without having to re-read and re-re-read.

Response: We have deleted the following acronyms; RPHC, NDoH, MDG, and MRC since they were used only twice in the manuscript. We reduced the use of some of the remaining acronyms by writing them up if possible.

Results - need some clarity. Seems very rosy. In other words, too subjective.

Response; Multiple data sources responded to the same questions on the acceptability of the outreach teams to the program staff and the community and on how the reach teams benefitted the program staff and the community. The findings and conclusions drawn are based on triangulated data from multiple data sources supported by the literature on outreach teams in the country and other settings. In addition, the results are supported with the quotations from the data sources.

Reviewer 2 Report

Review and comments:

1. Shorten the title to something more succinct – it is too long
2. The abstract needs to be more structures such as in an IMRAD format
3. Introduction:

a. Define “ward”
b. This section can be more concise and needs data about your patient care/community service infrastructure structure: numbers seen, number of providers, services, problems seen and in what frequencies, etc. Describe your patient population in terms of socio-economic status, education, etc. that reflects their needs for CHW and what the facilitators/barriers are.
c. CHWs are indispensable for outreach in certain types of community settings. Describe them: are they external workers, members of the community population, ages, etc. also what are numbers per capita patient population, special training etc.?

4. Materials Methods:

a. Describe your “contexts” – give an example. What you’ve written in 102+ is circular.
b. There is more overall representation of team leaders than any other group. How are other groups such as CHWs whom you spend a great deal of time describing represented in this sample? Ie why didn’t get more?
c. Who/how many community members were in the FGDs? How representative of the community health spectrum were they (Were they healthy, chronically ill, elderly, young, at-risk, etc)?
d. How do you know if you reached saturation (or even approached it on the last 2 groups)?
e. What was remuneration and/or other exchange for participation in the study?
f. Is the structure or the interview guide available?
g. Define acceptability and how it was communicated
h. What was the native language of the analysts? What was their understanding of local customs and culture?

5. Results/Discussion:

a. What are the workers getting paid? Is it sufficient for them as a part of satisfaction?
b. How do patients differ in terms of the complexity of their care? Is there a base level of complexity or chronicity of their health problems that represented in this sample?
c. Who are the “different stakeholders” in the community 369+. This is unclear.
d. What was your sampling bias in patient satisfaction 382+. Did you have any subjects who were not happy with the service or outreach they received?
e. How did you evaluate the final assessment/result? Did you present the results to the subjects to see if they agreed with your findings and assessments? There is no evidence of an evaluation of the validity of your results.
f. You state that you only looked at contextual factors, but you make a generalization that acceptability of the program translated into measurable benefits. How do you know it was solely due to contextual factors?

Overall, the presentation of the data (quotes) was satisfactory for a qualitative study, however there are weaknesses in the sampling and data collection with respect to heterogeneity of coverage of the different sectors of the healthcare community, which omits the influence of bias in reporting (there are an apparently higher percentage of managers in the data collection pool than there are patients or CHWs etc). For these, there is no checking/evaluation methodology (member checking , for example). You state triangulation but do not make an argument for what was done and how it demonstrated validity).

This paper is a start, but you need follow a format for qualitative research reporting that shows validity or at least vulnerabilities in your data collection procedures and the conclusions you draw from it.

Author Response

We have addressed all the comments to the best of our abilities.

Reviewer 3 Report

(1) Please provide a logical connection about the novelty of this work. It is difficult to justify publishing just because it has not been studied in the existing literature. Also, there is a lack of description of the research question and the focus of the research.

(2) In addition, please add a paragraph about the composition of the paper for each section in the last paragraph of the introduction.

(3) I recommend adding statistical analysis results in the results section. It seems difficult to guarantee the objectivity and generality of results only with qualitative analysis.

(4) The results and discussion do not seem to match well. Please discuss based on statistical analysis results and provide insights and implications.

(5) Please summarize the theoretical and practical contributions in Conclusion. In addition, please also add the limitations of this work and future directions of the research in this research stream.

Author Response

We thank the reviewer for the comments and suggestions on our manuscript. We have addressed the comments to the best of our abilities, in the document; the response is highlighted in a light blue font

Round 2

Reviewer 2 Report

Spell check. Add a legend for abbreviations used.

Author Response

We added a legend for the abbreviations that we used on page 16. We also proof read the manuscript for typos.